**Data Availability Statement:** All datasets used in the RNA-seq transcriptomic analysis are available

# Induction of systemic immunity through nasal-associated lymphoid tissue (NALT) of mice intranasally immunized with *Brucella abortus* malate dehydrogenase-loaded chitosan nanoparticles

**Soojin Shim**[1], **Sang Hee Soh**[1], **Young Bin Im**[1], **Choonghyun Ahn**[2], **Hong-Tae Park**[1], **Hyun-Eui Park**[1], **Woo Bin Park**[1], **Suji Kim**[1], **Han Sang Yoo**[1,3]*

1 Department of Infectious Diseases, College of Veterinary Medicine, Seoul National University, Seoul, South Korea, 2 Department of Biomedical Science, College of Medicine, Seoul National University, Seoul, South Korea, 3 BioMax/N-Bio Institute, Seoul National University, Seoul, South Korea

* yoohs@snu.ac.kr

## Abstract

Infection with *Brucella abortus* causes contagious zoonosis, brucellosis, and leads to abortion in animals and chronic illness in humans. Chitosan nanoparticles (CNs), biocompatible and nontoxic polymers, acts as a mucosal adjuvant. In our previous study, *B. abortus* malate dehydrogenase (Mdh) was loaded in CNs, and it induced high production of pro-inflammatory cytokines in THP-1 cells and systemic IgA in BALB/C mice. In this study, the time-series gene expression analysis of nasal-associated lymphoid tissue (NALT) was performed to identify the mechanism by which Mdh affect the target site of nasal immunization. We showed that intranasal immunization of CNs-Mdh reduced cell viability of epithelial cells and muscle cells at first 1 h, then induced cellular movement of immune cells such as granulocytes, neutrophils and lymphocytes at 6h, and activated IL-6 signaling pathway at 12h within NALT. These activation of immune cells also promoted signaling pathway for high-mobility group box 1 protein (HMGB1), followed by the maturation of DCs required for mucosal immunity. The CNs also triggered the response to other organism and inflammatory response, showing it is immune-enhancing adjuvant. The ELISA showed that significant production of specific IgA was detected in the fecal excretions and genital secretions from the CNs-Mdh-immunized group after 2 weeks-post immunization. Collectively, these results suggest that *B. abortus* Mdh-loaded CNs triggers activation of HMGB1, IL-6 and DCs maturation signaling within NALT and induce production of systemic IgG and IgA.

## 1. Introduction

Brucellosis is a highly contagious zoonotic disease caused by the genus *Brucella*, which is a group of gram-negative and facultative intracellular bacteria [1]. At present, live attenuated strains of *Brucella abortus*, such as Rev. 1, S19 and RB51, have been widely used for bovine

at Gene Expression Omnibus (GEO) (https://www.ncbi.nlm.nih.gov/geo/) under accession number GSE125765 (https://www.ncbi.nlm.nih.gov/geo/query/acc.cgi?acc=GSE125765).

**Funding:** This work was supported by the Strategic Initiative for Microbiomes in Agriculture and Food, Ministry of Agriculture, Food and Rural Affairs (No. 918020-4), Korea Health Industry Development Institute (KHIDI) (No. HI16C2130) and the Brain Korea (BK) 21 PLUS program for Creative Veterinary Science Research and the Research Institute of Veterinary Science, Seoul National University, Republic of Korea.

**Competing interests:** The authors have declared that no competing interest exist.

brucellosis vaccine [2]. However, these vaccines have some drawbacks, including antibiotic resistance, interference with diagnosis, abortion in pregnant animals and pathogenicity in humans [2]. Over the years, many efforts have been made to find safe and effective antigens including recombinant protein [1, 2].

Recent studies have reported that malate dehydrogenase (Mdh), a key enzyme in the tricarboxylic acid (TCA) cycle, is an important antigen in *B. abortus* infection of elk and cattle [3], and vaccination with Mdh promotes clearance of *B. abortus* infection in a mice model [4]. Additionally, Mdh was shown to be the most effective candidate for inducing pro-inflammatory immune responses in human leukemic monocyte cells (THP-1 cells) that were stimulated by several *B. abortus* recombinant proteins [5]. Similar to many pathogens, *Brucella* infections are usually transmitted through the mucosal membrane via oral or aerosol exposure [2, 6]. Therefore, the induction of mucosal immunity is important to build a primary barrier and prevent brucellosis. The induction of mucosal immunity in local site is able to stimulate both humoral and cell-mediated responses in mucosal and systemic sites [7].

To provoke mucosal immunity, an effective adjuvant and route of administration must be considered since the recombinant protein tends to be less immunogenic than the whole cell vaccine [8, 9]. Among the many adjuvants, chitosan nanoparticles (CNs) which are biocompatible and nontoxic have been shown to effective delivery vesicles to induce mucosal immunity [10, 11]. The chitosan, a natural linear polyaminosaccharide, is obtained by alkaline deacetylation of chitin and its positive charge by abundant amino groups reacts with negatively charged mucosal surfaces, as a useful polymer for mucosal delivery [10].

The inductive site of the mucosal immune response against inhaled antigen is known as the nasal-associated lymphoid tissue (NALT) in the upper respiratory tract (URT) [12]. The NALT is often compared to Waldeyer's ring of humans and has been considered as functionally equivalent to Peyer's patches in the gut [13]. NALT contains immunocompetent cell that play key roles in the defense against pathogens in upper respiratory tract and can induce various T helper cell subsets, including Th1, Th2 and Th17, in NALT [7, 13]. However, a transcriptomic regulation of nasal mucosa, the target site of nasal immunization, by intranasal immunization still remain unknown. In addition, it is not clear what characteristics of antigen can induce systemic immunity since systemic mucosal and humoral response is not always induced through mucosal immunization. Therefore, understanding of immune response in the NALT and following production of systemic antibody is crucial to facilitate the development of nasal vaccines.

Previously, our group loaded Mdh into the CNs and showed that Mdh is a promising antigen that elicits antigen-specific mucosal immune responses in BALB/c mice [14]. We assumed that appropriate activation of immune response of nasal cavity by composition of Mdh and CNs induced systemic immunity. Therefore, in the present study, the transcriptional responses of NALT were analyzed to identify the mechanism by which Mdh affect the target site of nasal immunization. The transcriptomic regulation within NALT was evaluated by identifying differentially expressed genes (DEGs) of NALT from mice intranasally immunized with CNs-Mdh. In addition to the transcriptomic analysis, induction of mucosal and systemic humoral immunity was investigated after 2 weeks-post immunization. The study will be beneficial for further understanding of the immune responses against Mdh, initiative biological process of NALT following intranasal immunization and the rational design of vaccination strategies.

## 2. Materials and methods

### 2.1. Preparation of Mdh

The Mdh was prepared according to a previously described method [14]. Briefly, *E. coli* strain harboring a recombinant plasmid constructed in a previous study [15] was grown overnight at

37˚ C in LB medium (Duchefa, Netherlands) and protein expression was induced by adding isopropyl β-d-1-thiogalactopyranoside (IPTG, Amresco, USA) at a final concentration of 0.3 mM. Then, the soluble protein fraction was purified using a His SpinTrap (GE Healthcare, UK) column according to the manufacturer's instructions after the pellet was sonicated at 10,000 Hz on ice. The proteins were constructed with a pCold vector containing a chaperone trigger factor (TF) to produce soluble and functional proteins. Therefore, as a vector control, TF was also purified using the above method. The endotoxin content of proteins was confirmed with a Pierce Chromogenic Endotoxin Quant kit (ThermoFisher, USA). The purity of proteins was confirmed using SDS-PAGE and Western blotting as previously described [15]. The concentration of purified proteins was measured by Pierce BCA protein assay kit (Pierce, USA) following the manufacturer's instructions and stored at −20˚C until use.

## 2.2. Preparation of CNs-Mdh

To prepare CNs, an ionic gelation method was used according to a previously developed method [10, 14]. Briefly, 0.125 g of water-soluble chitosan (Jakwang. Ansung, Korea) was dissolved in 25 ml of distilled water (0.5 w/v%) and mixed with 150 μl of 8% Tween 80 (Sigma, USA) solution. A total of 250 μl of TPP (10 w/v%) was dropped into 25 ml of 0.5 wt% chitosan solution through syringe under magnetic stirring and sonicated on ice (6 W, 10 min). The CNs were obtained by centrifugation for 10 min at 12,000 rpm and freeze-dried.

To load Mdh and TF into the CNs, one milliliter of protein (3 mg/ml) containing 20 mg of CNs was kept at 37˚C for 24 h in a shaking incubator. The supernatants containing free protein were removed by centrifugation (10 min, 8,000 rpm), and protein-loaded CNs were then freeze-dried. After loading antigens, the morphology of the particles was assessed using field emission SEM (FESEM, SUPRA 55VP; Carl Zeiss, Germany). The samples were placed on a cupper grid and air-dried at RT. Then, the samples were coated with platinum prior to examination by FESEM. The particle size was measured by DLS using a zeta potential and particle size analyzer (ELS Z-1000; Otsuka Electronics, Japan). The loading efficiency (LE) in the CNs was determined by calculating the free protein using a BCA protein assay (Pierce, USA) and the following equation:

$$LE(\%) = \frac{\text{Total amount of the protein} - \text{Free protein}}{\text{Total amount of the protein}} \times 100$$

## 2.3. Ethics statement

The 6-week-old BALB/c female mice were purchased from Orient Bio (Orient Bio Inc., Seongnam, Korea). The study procedures were reviewed and approved by the Institutional Animal Care and Use Committee (IACUC) at Seoul National University (IACUC approval number: SNU-180403-7) according to the principles established by the Animal Protection Act and the Laboratory Animal Act in Republic of Korea. All animal surgeries were performed with highly minimized pain and suffering.

## 2.4. Nasal immunization and isolation of NALT

Each mouse was immunized by nasal route with 30 μg of purified TF and *B. abortus* Mdh with CNs in 20 μl of PBS, CNs alone or PBS. After 1 h, 6 h, and 12 h of immunization, NALT was obtained from the immunized mice (n = 3) as described previously [16]. Briefly, mice were sacrificed by ether asphyxia and the lower jaws were removed. Then, the palates were cut and removed. The paired NALT structures from each side of the septum were gently removed with fine dissection forceps.

## 2.5. RNA-seq analysis

Samples of NALT from CNs-Mdh immunized mice at 1 h, 6 h, and 12 h were chosen for RNA-sequencing analysis. As controls, in parallel time series, samples from the PBS-, CNs-, CNs-TF-immunized groups were also prepared. Total RNA was then extracted from the tissue using a RNeasy Mini kit (Qiagen, Germany). Subsequent RNA preparation steps were carried out at the TheragenEtex Bio Institute (Seoul, Korea). The purity and integrity of RNA were determined by denaturing gel electrophoresis, OD260/OD280 ratio, and analysis with an Agilent 2100 Bioanalyzer (Agilent Technologies, CA, USA). The samples were then used to generate sequencing libraries with a TruSeq Stranded mRNA Sample Preparation Kit (Illumina, CA, USA) and sequenced on an Illumina HiSeq 2500 sequencer following the manufacturer's instructions. Low-quality reads were filtered, and the reads were mapped to the reference genome related to the species using the TopHat aligner [17]. The gene expression level was measured with Cufflinks v2.1.1 [18] using the gene annotation database of the species. To improve the accuracy of the measurement, multi-read-correction and frag-bias-correction options were applied. All other options were set to default values.

## 2.6. Transcriptomic data analysis

The RNA-seq gene lists of differentially expressed genes (DEGs) obtained at each time point and differences were considered significant if $p \leq 0.05$ and a fold-change $> |2|$ were obtained. The fold change of "± inf" (infinity) means that expression was only detected in one immunized group and it was considered significant if $p \leq 0.05$. Gene-enrichment and functional annotation analysis for significant probes was performed using Gene Ontology (http://geneontology.org) and PANTHER (http://pantherdb.org/about.jsp). Additionally, the canonical pathways and functional analyses of the DEGs were generated through the use of IPA (QIAGEN Inc., https://www.qiagenbioinformatics.com/products/ingenuity-pathway-analysis) [19]. All datasets used in the RNA-seq transcriptomic analysis are available at Gene Expression Omnibus (GEO) (https://www.ncbi.nlm.nih.gov/geo/) under accession number GSE125765 (https://www.ncbi.nlm.nih.gov/geo/query/acc.cgi?acc=GSE125765).

## 2.7. Quantitative PCR analysis

To verify the RNA-seq analysis data, six genes (*Il1rn*, *Batf*, *Tlr4*, *Il6*, *Il23a* and *Tnf*) associated with innate immune response were selected and subjected to quantitative real-time PCR (qRT-PCR). Briefly, mRNA was reverse-transcribed using a Quantitect Nova Reverse Transcription Kit (Qiagen, Germany). Real-time PCR was performed with 2 μl of cDNA using the Rotor-Gene SYBR Green PCR Kit (Qiagen) and Rotor-Gene Q real-time PCR cycler (Qiagen). The cycling parameters were as follows: 95˚C for 5 min for one cycle followed by 45 cycles of 95˚C for 15 sec and 60˚C for 45 sec. The *Gapdh* gene was used as an internal control. The primers used are listed in S1 Table. Relative expression was calculated using the comparative CT ($2^{-\Delta\Delta CT}$) method. Additionally, the Pearson correlation coefficient was calculated in Microsoft Excel.

## 2.8. Sampling

At 2 wpi, serum samples, nasal washes, genital secretions and fecal excretions were obtained from the mice (n = 5). Briefly, blood samples were collected from the heart and centrifuged for sera preparation (1000 × g, 10 min at 4˚C). The nasal washes were collected by injection of 300 μl PBS through the trachea towards the nose and centrifugation (1,000 × g, 10 min at 4˚C). For genital secretion samples, the genitals were harvested and chopped in 1 ml of PBS, and the

supernatants were collected by centrifugation (16,000 × g, 5 min at 4˚C). The fecal samples were weighed, dissolved in to PBS (1 w/v%), and centrifuged (16,000 × g, 5 min at 4˚C). All of the PBS used for sample collection included a protease inhibitor cocktail (Sigma, USA) and the samples were stored at − 70˚C until use.

## 2.9. Detection of IgG, IgG1, IgG2a, and IgA titers

To measure the IgG, IgG1, IgG2a, and IgA titers, an indirect ELISA was performed using obtained samples. For TF-untagged antigen preparation, the recombinant proteins were cleaved with HRV 3C protease (TaKaRa Bio, JAPAN) at 4˚C for 16 h. After SDS-PAGE, the pure proteins without TF, HRV 3C protease, and noncleaved proteins were acquired using a Gel Extraction Kit (Koma Biotech Inc, Seoul, Korea). 96-well microplates were coated by incubating purified recombinant proteins (10 ng/well) in a coating buffer (14.2 mM sodium carbonate, 34.9 mM sodium bicarbonate, and 3.1 mM sodium azide, pH 9.6) overnight at 4˚C for an ELISA. The plates were blocked with 1% bovine serum albumin (BSA; bioWORLD, OH, USA) in a solution of phosphate-buffered saline (PBS) containing 0.05% Tween 20 (PBST) for 2 h at 37˚C. The sera were diluted 1:4000 to detect specific IgG, IgG1 and IgG2a and IgA. For specific IgA detection from wash samples, the nasal wash samples were diluted 1:80, the genital secretions were diluted 1:800, and the fecal excretions were diluted 1:25. After washing with PBST, diluted serum samples were added to the wells and incubated for 2 h at RT. Horseradish peroxidase (HRP)-conjugated goat anti-mouse IgG (Bio-Rad Laboratories, USA), IgG1 (Southern Biotechnology, USA), IgG2a (Southern Biotechnology, USA), and IgA (BioFX Laboratories, USA) diluted in PBS containing 1% BSA were used to detect IgG, IgG1, IgG2a, and IgA, respectively. Total IgA titers in the serum samples, nasal washes, genital secretions and fecal excretions were measured using a Mouse IgA ELISA Ready-SET-Go! Kit (eBioscience, San Diego, CA) according to the manufacturer's instructions. The nasal wash samples were diluted 1:80, the genital secretions were diluted 1:800, the fecal excretions were diluted 1:25, and the sera were diluted 1:4000. Color development was performed by adding the substrate 3,3′,5,5′-tetramethyl-benzidine (TMB; Sigma). The absorbance was measured at 450 nm using a VersaMax microplate reader (Molecular Devices Corporation, CA, USA).

## 2.10. Statistical analysis

All results are expressed as the mean ± standard deviation (SD). Differences were considered statistically significant at a *p-value* < 0.05. Differences in titer of IgA were analyzed with one-way analysis of variance (ANOVA) and Tukey's test for post-hoc analysis. All experiments were repeated at least 3 times. Statistical analyses were performed by GraphPad Prism 5 software (GraphPad Software, Inc., La Jolla, CA, USA).

# 3. Results

## 3.1. Preparation of Mdh and CNs-Mdh

To produce *B. abortus* Mdh, fusion proteins were constructed with a pCold vector containing a chaperone trigger factor (TF) commonly used to produce soluble and functional proteins. As a vector control, TF was also purified. As shown in Fig 1A, the protein sizes of the TF and Mdh were approximately 52 kDa and 85.71 kDa, respectively. Western blotting was also performed using an anti-6X Histag antibody to confirm these proteins. To enhance the immunogenicity and control the release of Mdh, CNs were prepared as adjuvants using the tripolyphosphate (TPP)-based ionic gelation method. The morphology of the particles was assessed using field emission scanning electron microscopy (SEM) and Fig 1B shows that the CNs were spherical

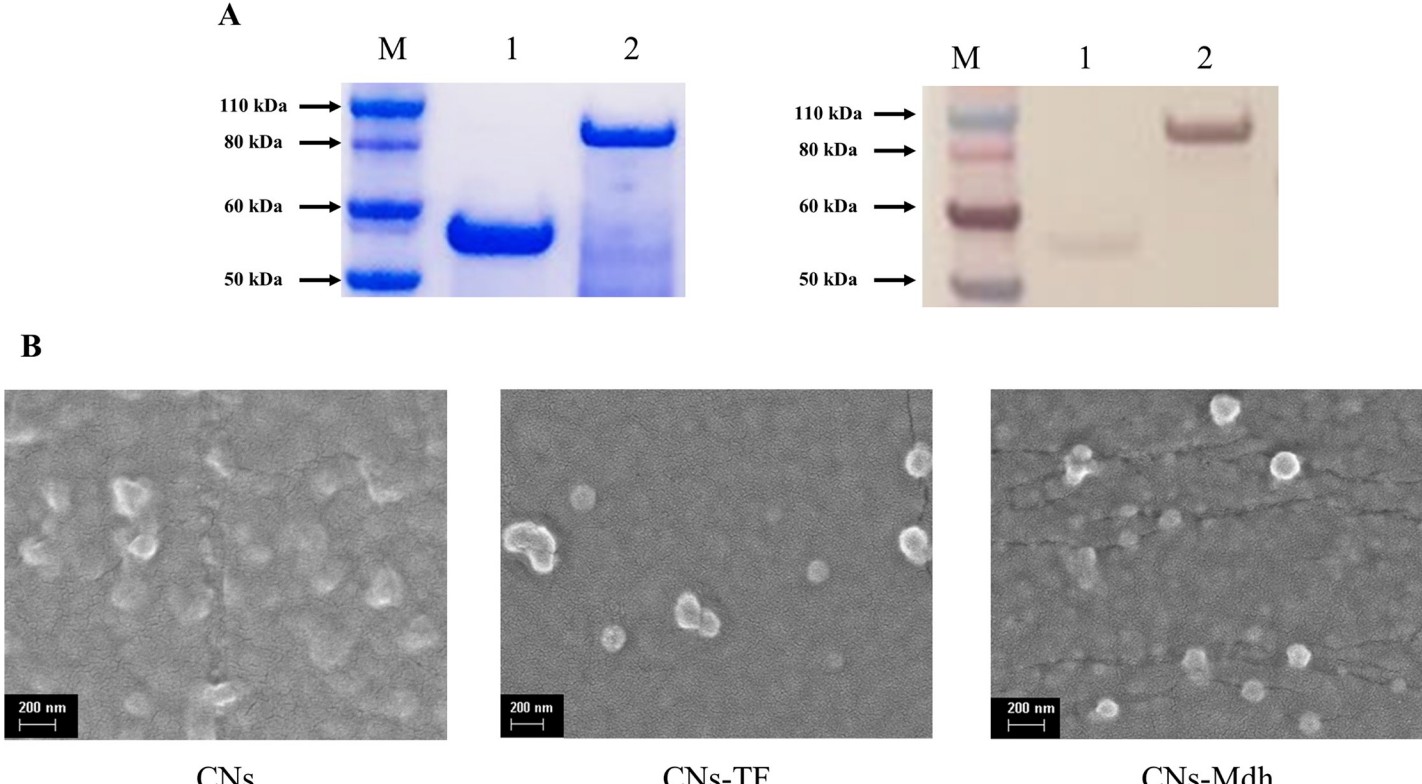

**Fig 1. The confirmation of recombinant *B. abortus* proteins.** SDS-PAGE and Western blotting (A). Lane M: Molecular size standard, 1: Control, TF (53 kDa), 2: Mdh (92.71 kDa). SEM photographs of prepared CNs (B).

and less than 200 nm in size. After loading antigens, the sizes of the CNs-TF and CNs-Mdh measured by dynamic light scattering (DLS) were 217.8±45.2 nm and 405.0±94 nm, respectively. The LE was 51.45 ± 3.48% and 78.22 ± 0.75%, which means that 1.54 mg of TF and 2.34 mg of Mdh was loaded per 20 mg of CNs.

### 3.2. Gene expression

To observe the immune responses induced by antigen and adjuvant, the transcriptome data of the NALT taken 1, 6, and 12 h post-immunization (hpi) with CNs-Mdh were compared to three controls, PBS-, CNs- and TF-immunized group (Fig 2A). The differentially expressed genes (DEGs) were obtained at each time point and the criteria of $p \leq 0.05$ and a fold-change $> |2|$ were chosen to determine significantly up-regulated or down-regulated genes following immunization. Genes commonly regulated between CNs vs PBS (control), CNs-Mdh vs PBS (control) and CNs-TF vs PBS (control) were used to evaluate the effect of CNs. Next, the effect of Mdh was evaluated using CNs-Mdh vs CNs (control) and CNs-TF vs CNs DEGs. Finally, CNs-Mdh vs CNs-TF (control) DEGs were used to identify the effect of Mdh compared to the vector control. The number of DEGs according to time is shown in Fig 2B.

### 3.3. The effect of CNs

DEGs in the CNs vs PBS, CNs-Mdh vs PBS and CNs-TF vs PBS set are shown in Venn diagram (Fig 3A). The number of commonly up-regulated genes was 8 (1 h), 161 (6 h) and 17 (12 h). The number of commonly down-regulated genes was 6 (1 h), 10 (6 h) and 6 (12 h),

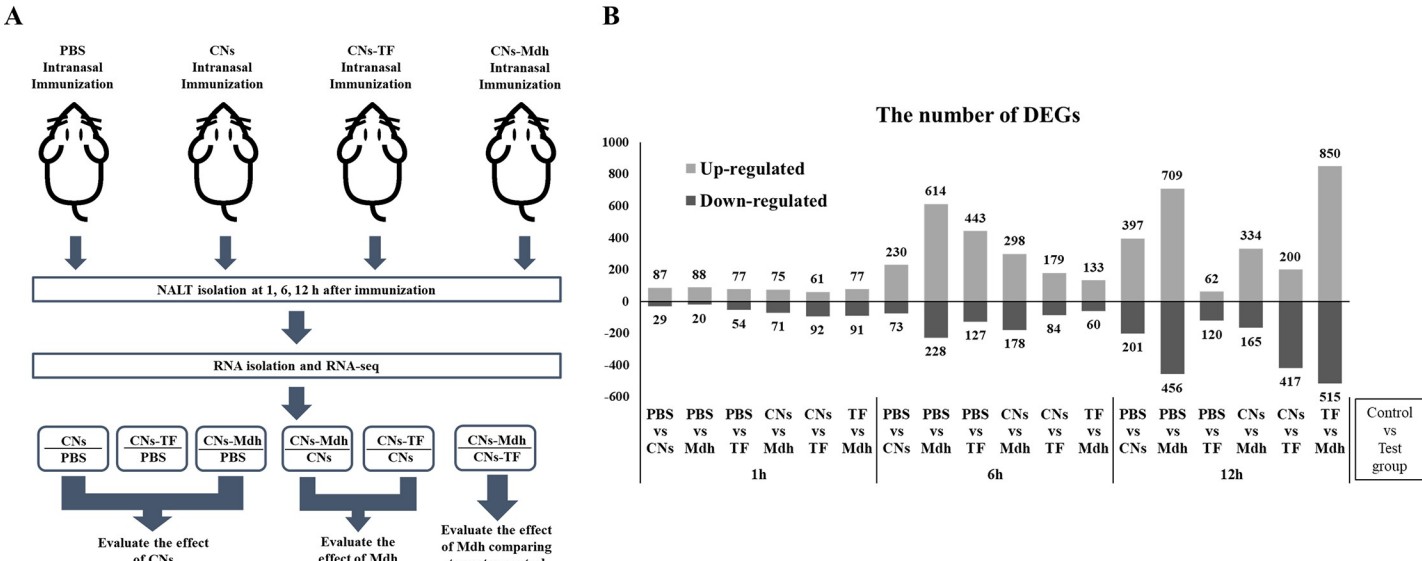

**Fig 2.** The experimental design for the analysis of transcriptomic profiling (A). The number of DEGs in each compared subset (B). The differentially expressed genes (DEGs) were obtained at each time point and the criteria of p ≤ 0.05 and a fold-change > |2| were chosen to determine significantly up-regulated or down-regulated genes following immunization.

respectively. Gene ontology (GO) enrichment analysis was carried out by PANTHER bioinformatics tools, using these genes. The top 10 GO terms are summarized in Fig 3B. Significantly matched terms in the 1 h data were not detected, showing that GO between CNs- and CNs-Mdh immunized group at 1 hpi were not significantly different. All of the top 10 terms from the 6 hpi and 12 hpi samples were up-regulated terms due to the low enrichment of the down-regulated terms. The results showed that response to other organism became active (GO:0009605; GO:0009607; GO:0043207; GO:0051707) and control of immune response including defense (GO:0006952), response to cytokine (GO:0071345; GO:0034097) and inflammatory response (GO:0006954; GO:0002376; GO:0002682) was matched significantly at 6 hpi. The expression patterns of these genes suggest that antigen recognition progressed from 1 hpi to 6 hpi after administration using CNs. At 12 hpi, the expression levels of genes for migration of neutrophil, granulocyte and leukocyte were matched significantly. In addition, activation of immune response such as complement activation, fever generation and cytokine production was observed at 12 hpi.

### 3.4. The immunogenicity of Mdh

To identify the immunogenicity of Mdh, DEGs in the CNs-Mdh vs CNs and CNs-TF vs CNs set are compared and Venn diagram was shown in S1A Fig. The number of commonly up-regulated genes was 16 (1 h), 69 (6 h) and 17 (12 h), respectively. The number of commonly down-regulated genes was 20 (1 h), 39 (6 h) and 37 (12 h), respectively. The GO terms analysed by GO enrichment analysis using these genes were shown in S1B Fig. The results showed that development associated with tissue, odontogenesis and epithelial cells was matched, but it showed low fold enrichment. Then, at 12 hpi, response to other organism (GO:0009607; GO:0006950; GO:0009605; GO:0043207) and defense response which were similarly observed at GO analysis of common DEGs at 6 hpi from subset CNs vs PBS, CNs-Mdh vs PBS and CNs-TF vs PBS were matched significantly at 12 hpi.

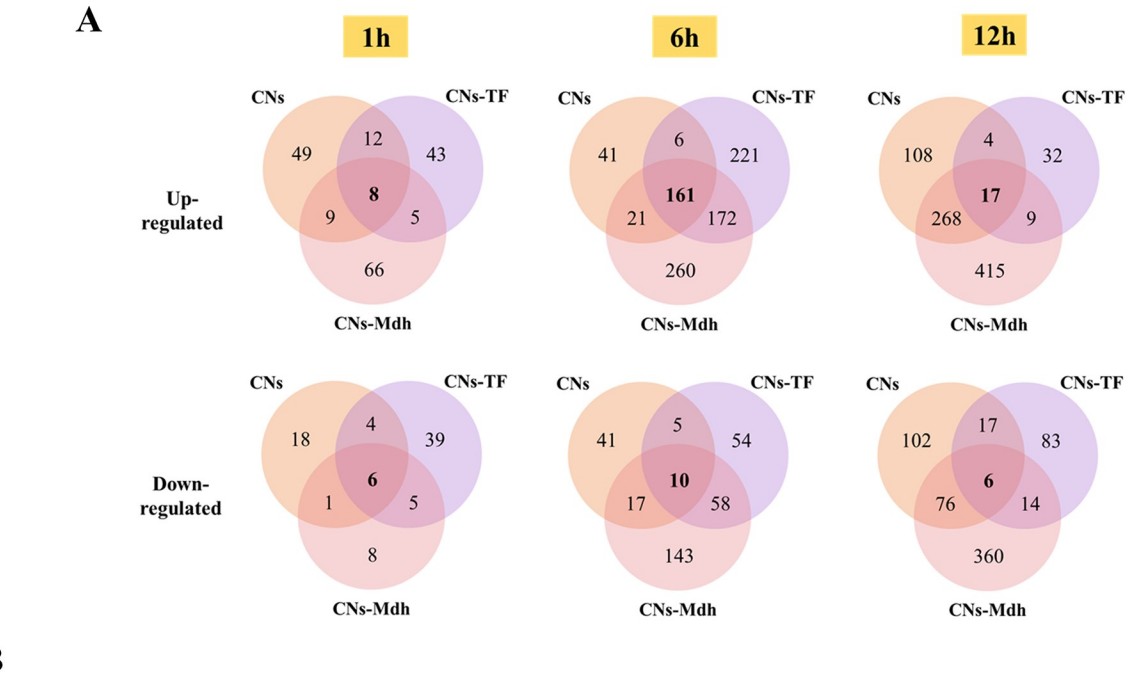

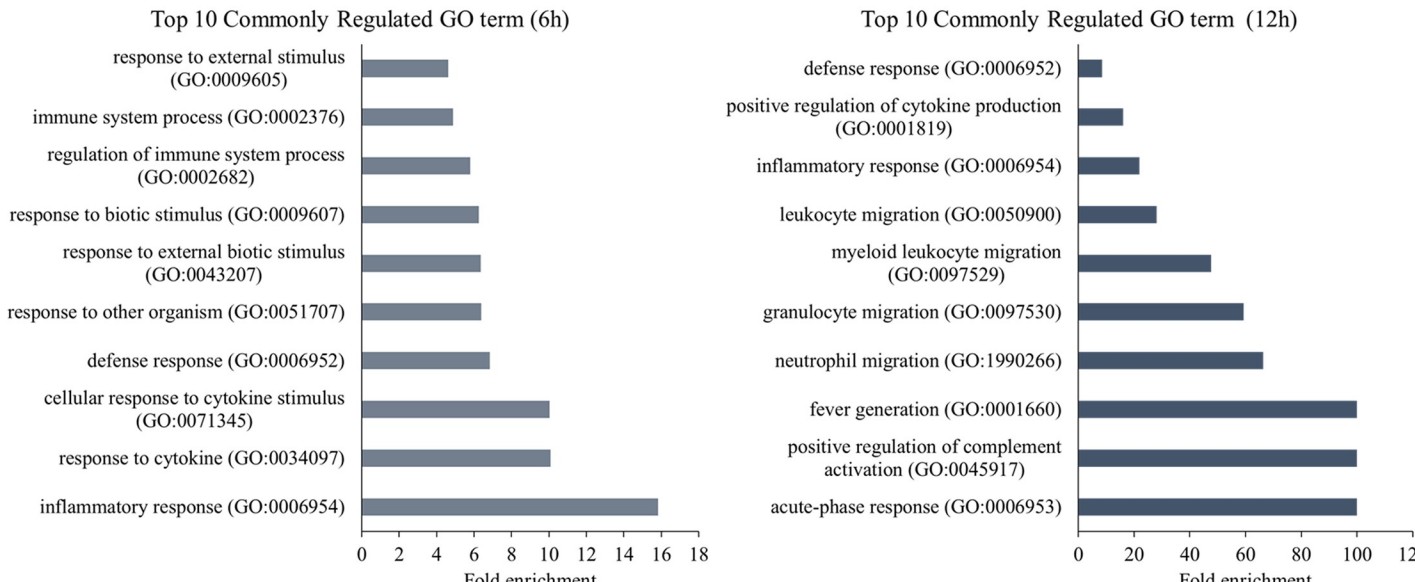

**Fig 3. Genes and functions that commonly regulated between CNs vs PBS, CNs-TF vs PBS and CNs-Mdh vs PBS.** Venn diagram of DEGs from only the CNs-, CNs-TF and CN-Mdh-immunized groups compared to the PBS-immunized group (A). Gene Ontology enrichment analysis using commonly regulated genes from only the CNs-, CNs-TF and CN-Mdh-immunized groups (B).

As compared to CNs-TF- and CNs-immunized group, the number of up-regulated genes in only from CNs-Mdh-immunized group was 59 (1 h), 229 (6 h) and 214 (12 h), respectively. The number of down-regulated genes was 51 (1 h), 139 (6 h) and 128 (12 h), respectively. To determine the specific and major 'Diseases and function' affected by CNs-Mdh, the functional annotation of the DEGs using these genes according to the *p-value* and Z-score (a score of the predicted direction of the pathway) was conducted and top 10 functions were summarized in the Table 1. It showed that genes associated with inflammation of the organs and body cavity

were up-regulated at 1 hpi while genes associated with "cell viability" including cell cycle progression, neurons, epithelial cells and muscle cells were down-regulated. Genes related in "differentiation and development of connective tissue cells" were also down-regulated, showing that inflammation responses were induced more than CNs- and CNs-TF-immunized group. Then, at 6 hpi, the functions of "Cell movement", "Homing of cells", and "T cell migration" were up-regulated. The function of "Cell viability" was up-regulated at 6 hpi while it was down-regulated at 1 hpi. In addition to the inflammatory response shown at 6 hpi, active movements of leukocytes, phagocytes, granulocytes were observed at 12 hpi, indicating that the inflammatory response was induced.

## 3.5. The activated signaling pathways

The canonical pathway analysis of DEG subsets from CNs-Mdh vs CNs-TF was performed using Ingenuity Pathway Analysis (IPA) software. The top 20 pathways according to the *p-value* and Z-score (a score of the predicted direction of the pathway) are included in Fig 4. These pathways had a high level of statistical significance; the -log p was higher than 5.0 (Fig 4A) and the Z score (Fig 4B) was higher than 2.0 at the 12 hpi. Signaling pathway analysis revealed that "Dendritic cell maturation", "High mobility group box 1 (HMGB1) Signaling", "Interleukin (IL)-6 Signaling", "Acute phase Signaling", "Triggering receptor expressed on myeloid cells 1 (TREM1) Signaling", "Production of Nitric Oxide and Reactive Oxygen Species in Macrophages", "p38 MAPK Signaling", "NF-κB Signaling" and "Toll-like Receptor Signaling" pathways were predicted to be activated significantly (both *p* value and Z score) after immunization with CNs-Mdh. Among them, "HMGB1 Signaling" and "IL-6 Signaling" pathways represent overall structure of these key signaling pathways (Fig 5). As shown in Fig 5A, activation of TLR signaling (by TLR4) and inflammatory cells (by *IL-1*, *TNF-α* and *IFN-γ*) activated the NF-κB and MAPK pathways in the HMGB1 pathway. Innate cell activation by HMGB1 that induces pro-inflammatory cytokines including TNF-α, IL-1α, IL-1β, IL-6 and IL-8 plays a role in mediating dendritic cells (DCs) maturation and activation. As a pro-inflammatory cytokine, IL-6 signaling pathway was predicted to be significantly activated through the increased expression of *Il1b*, *Il6*, *NF-kB*, *NF-IL6*, *Stat3* and *SOCS3* (Fig 5B).

## 3.6. Activated gene network for mucosal immunity

To predict activation of mucosal immune response and induction of IgA [20–23] following intranasal immunization of CNs-Mdh, significant gene networks were analyzed (Fig 6). The color of each node indicates its Fc value. According to the gene network analysis, genes related in activation of lymphocytes, chemotaxis and cell movement of DCs were actively expressed and expression of *Il1a*, *Il1b*, *Tnf*, *Vip* and *Csf2rb* connected these networks (12 hpi). Additionally, genes involved in synthesis of retinoic acid (*Aldh1a3*), modulators for T cell and DC function (*Csf*, *Csf1*, and *Csf3*), generation of IgA-ASCs and IgA (*Tgfb1*, *Tgfbi*, and *Cd40*) were expressed concomitantly.

To confirm the production of IgA, specific IgA and total IgA were detected in sera, nasal wash, fecal excretions and genital secretions after 2 wpi. In addition, specific IgG levels were measured for sera. For specific IgA (Fig 7A), significant production of IgA was detected in the fecal excretions and genital secretions from the CNs-Mdh-immunized group compared to CNs-TF-immunized group. However, no significant differences were observed in the nasal wash and serum IgA levels between the CNs-TF and CNs-Mdh-immunized groups. In addition to specific IgA, the CNs-Mdh elicited a significantly higher titer of specific IgG than CNs-TF-immunized group which served as the vector control (Fig 7B). The main subtype produced after immunization was IgG1, while the titer of IgG2a was not significantly enhanced following

**Table 1. The functional annotation of DEGs from CNs-Mdh vs CNs and vs CNs-TF.**

|  | Diseases or Functions Annotation | p-Value | Predicted Activation State | Activation z-score |
|---|---|---|---|---|
| 1h | Inflammation of body cavity | 0.000267 | Increased | 2.64 |
|  | Inflammation of organ | 5.22E-05 | Increased | 2.333 |
|  | Organismal death | 0.000603 | Increased | 2.012 |
|  | Cell viability of neurons | 0.0000419 | Decreased | -2.569 |
|  | Cell viability | 0.000218 | Decreased | -2.925 |
|  | Cell cycle progression | 0.00165 | Decreased | -2.352 |
|  | Differentiation of connective tissue cells | 5.95E-05 | Decreased | -2.329 |
|  | Development of connective tissue cells | 1.77E-07 | Decreased | -2.27 |
|  | Proliferation of epithelial cell lines | 0.000209 | Decreased | -2.415 |
|  | Migration of smooth muscle cells | 7.71E-05 | Decreased | -2.198 |
| 6h | Cell movement | 1.09E-19 | Increased | 4.968 |
|  | Homing of cells | 1.27E-21 | Increased | 4.57 |
|  | Leukocyte migration | 5.14E-22 | Increased | 4.567 |
|  | Migration of cells | 3.05E-19 | Increased | 4.439 |
|  | Recruitment of phagocytes | 2.83E-18 | Increased | 4.278 |
|  | Chemotaxis | 1.08E-19 | Increased | 4.19 |
|  | Recruitment of leukocytes | 9.08E-19 | Increased | 4.071 |
|  | Cell movement of mononuclear leukocytes | 8.70E-10 | Increased | 4.045 |
|  | T cell migration | 3.11E-07 | Increased | 3.999 |
|  | Cell viability | 2.38E-10 | Increased | 3.982 |
| 12h | Leukocyte migration | 7.51E-46 | Increased | 6.18 |
|  | Cell movement of leukocytes | 9.91E-41 | Increased | 5.94 |
|  | Cell movement of phagocytes | 9.91E-39 | Increased | 5.748 |
|  | Homing of leukocytes | 1.47E-32 | Increased | 5.744 |
|  | Chemotaxis of leukocytes | 3.50E-31 | Increased | 5.498 |
|  | Cell movement of mononuclear leukocytes | 1.17E-19 | Increased | 5.183 |
|  | Cell movement of granulocytes | 4.51E-42 | Increased | 5.151 |
|  | Inflammatory response | 3.86E-47 | Increased | 5.134 |
|  | Adhesion of immune cells | 1.21E-33 | Increased | 5.097 |
|  | Chemotaxis of phagocytes | 6.16E-30 | Increased | 5.08 |

immunization. For total IgA (Fig 7C), titers of IgA in genital secretions and fecal excretions were significantly increased in CNs-Mdh-immunized group compared to those following the CNs-TF-immunization. In the nasal washes, a significantly high titer of IgA in the CNs-Mdh-immunized group was measured, compared to the PBS- and CNs-immunized groups. However, there were no significant differences in sera.

## 3.7. Validation of RNA-seq data

To verify the RNA-seq results, qRT-PCR was conducted using the same experimental RNA samples with six genes (*Il1rn*, *Batf*, *Tlr4*, *Il6*, *Il23a* and *Tnf*) associated with the innate immune response including the IL-6 signaling, HMGB1 signaling and DCs maturation. The correlation coefficient of the expression values from the RNA-seq and qRT-PCR data was 0.92048 (S2 Fig).

## 4. Discussion

Over the years, many studies have investigated the immunogenic antigens of *Brucella abortus*. Among these antigens, Mdh have been shown to be promising antigen candidate for a *Brucella*

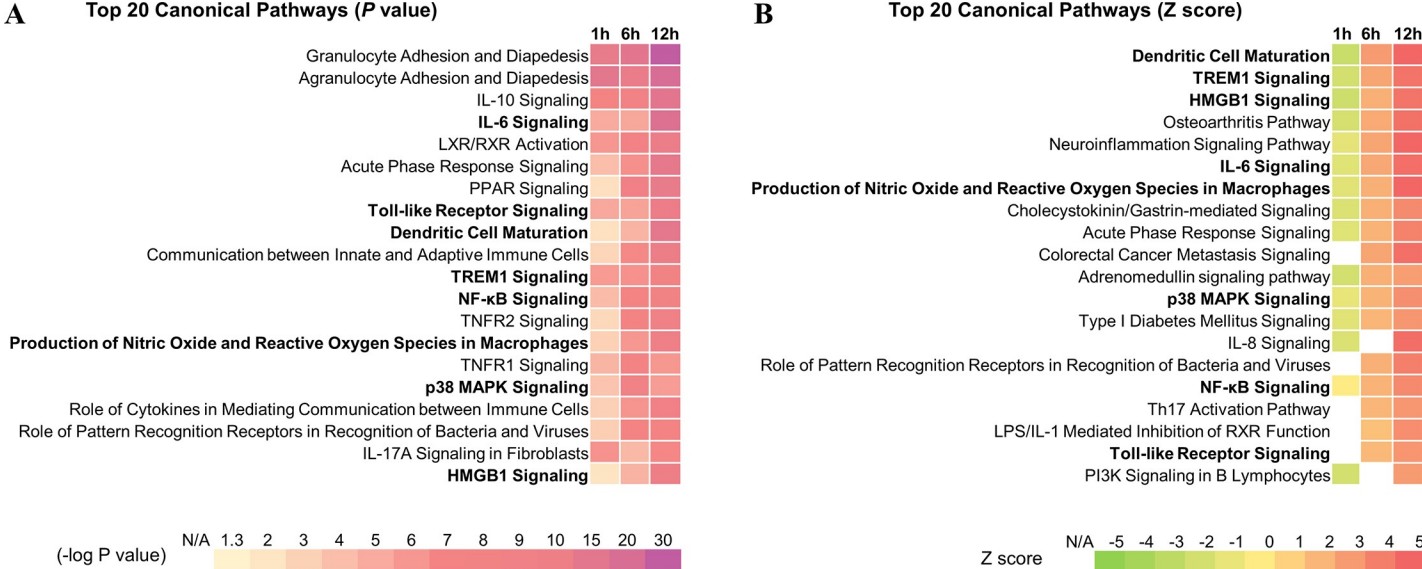

**Fig 4. Pathway and functional analyses were performed using the DEG subset from CNs-Mdh vs CNs-TF.** The top 20 canonical pathways as per *P-value* and Z score. The canonical pathway analyses and function annotations were generated through the use of IPA (QIAGEN Inc., https://www.qiagenbioinformatics.com/products/ingenuity-pathway-analysis) [19].

spp. vaccine both *in vitro* and *in vivo* in our previous studies [5, 14]. In the present study, *B. abortus* Mdh were loaded in the mucoadhesive adjuvant, CNs, and CNs-Mdh were intranasally immunized into the mice. The transcriptomic analysis of the NALT from CNs-Mdh-

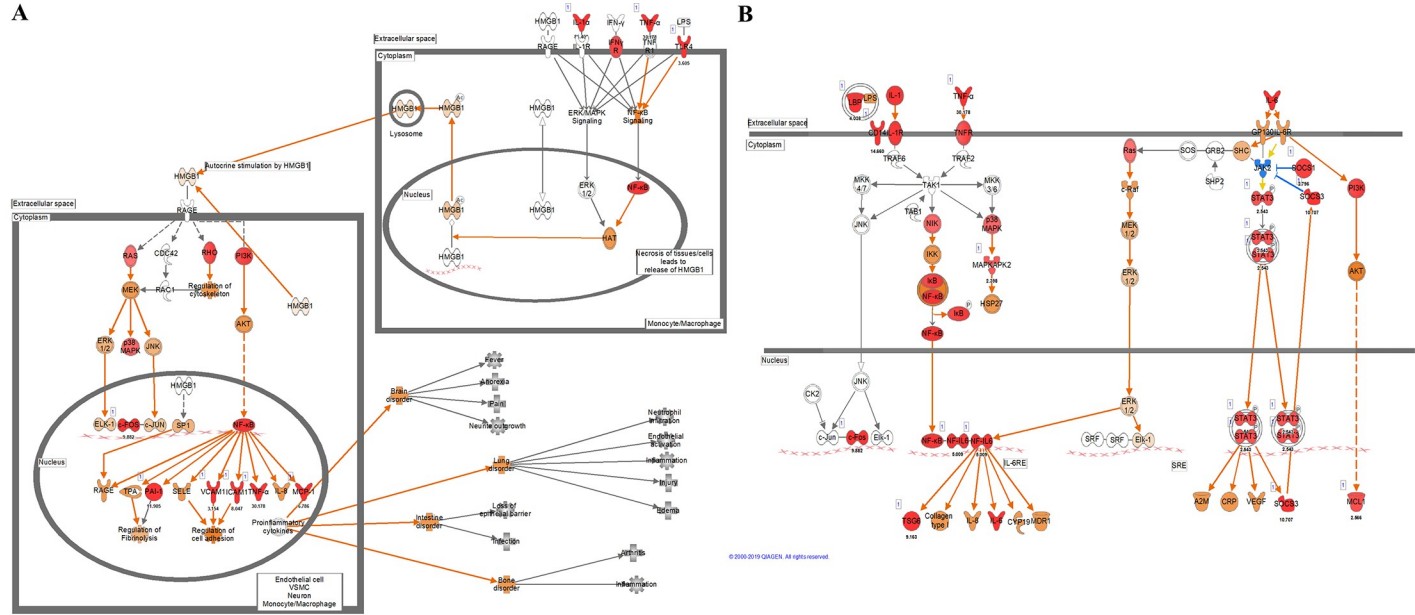

**Fig 5. Canonical pathways analyzed using the DEG subset from CNs-Mdh vs CNs-TF.** The significant canonical pathways of ""HMGB1 Signaling" (A) and "IL-6 Signaling" pathways (B) represent overall structure of these key signaling pathways by intranasal immunization of CNs-Mdh. These pathways were analyzed by Ingenuity Molecule Activity Predictor (MAP). The red nodes indicate genes significantly up-regulated, and the orange nodes indicate genes predicted to be activated. The intensity of color corresponds to an increase in fold change levels. The canonical pathway and upstream regulators analyses were generated through the use of IPA (QIAGEN Inc., https://www.qiagenbioinformatics.com/products/ingenuity-pathway-analysis) [19].

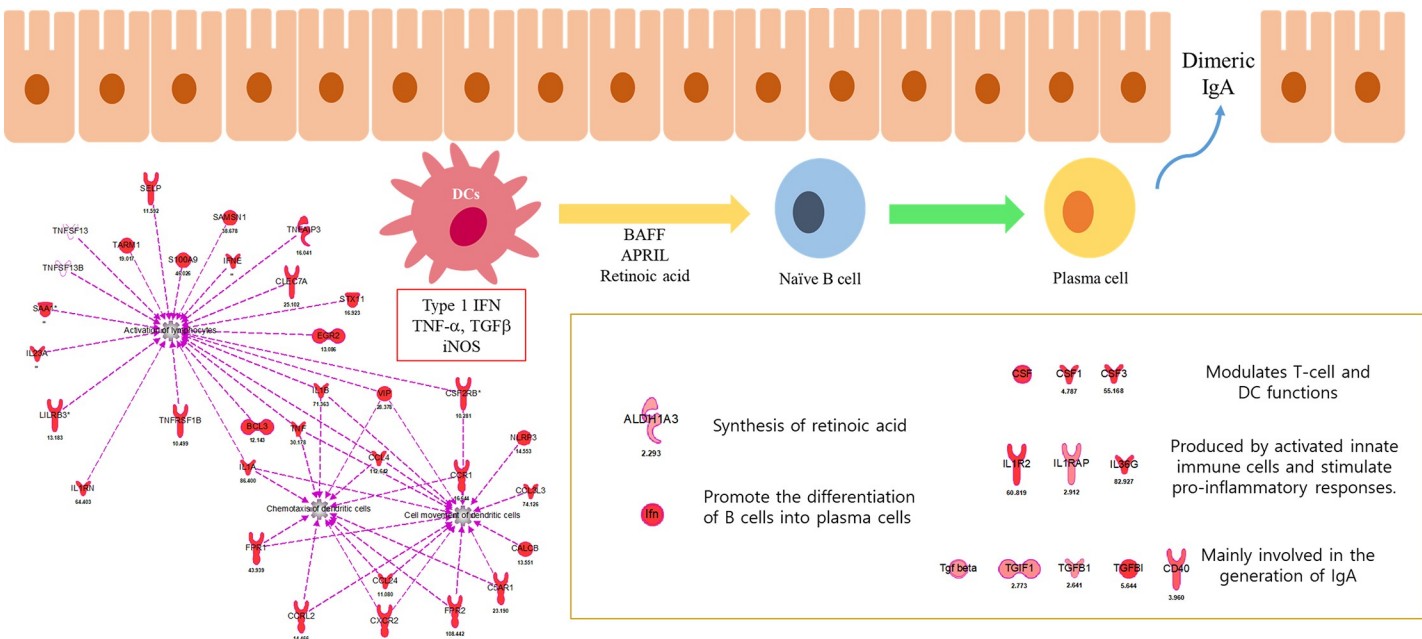

**Fig 6. Gene networks involved in DCs maturation and mucosal ASCs in NALT.** According to bioinformatics analysis, networks for DCs antigen presentation was induced by activation of lymphocytes and genes related with mucosal ASCs were also highly expressed. The gene networks analyses were generated through the use of IPA (QIAGEN Inc., https://www.qiagenbioinformatics.com/products/ingenuity-pathway-analysis) [19].

immunized mice was performed with time-series data to examine the responses of nasal cavity, the target site of nasal immunization. In addition, the induction of mucosal and systemic humoral immunity was investigated.

As valuable polymers, CNs have been demonstrated due to their ability to enhance mucosal absorption, mucosal adhesion, and control the release of antigens [11]. In this study, the spherical features of the prepared CNs were confirmed by SEM. The DLS results showed that protein-loaded CNs were larger after loading of the antigens. As many studies have shown, particle size is critical for absorption in the mucosal membrane and avoiding clearance in the airways [24]. Recent studies have demonstrated that M cells participate in antigen uptake from the nasal cavity to the NALT [25, 26]. The M cells from the intestine have been reported to transport antigen to the Peyer's patches, the local lymphoid tissue in the gut [26]. The preferred particle size for uptake by the M cells is < 1 μm [27]. The size of the nanoparticles in the present study was < 1μm, and it is assumed that CNs-Mdh could be transported to the NALT.

When antigen is inhaled, pattern recognition receptors (PRRs) on antigen-presenting cells (APCs), such as dendritic cells (DCs) and macrophages determine the origin of the antigens [28]. This antigen presentation leads to antibody-secreting cells (ASCs). Adaptive immunity is known to be controlled by PRRs-induced signals, which instruct when and how to respond to a particular infection [28, 29]. At the 6 hpi, group of CNs-Mdh, CNs-TF and CNs alone, the expressions of common genes involved in response to other organism were increased in NALT. Our results indicate that the CNs were recognized by PRRs on the immune cells during the 1 hpi to 6 hpi, inducing defense and cytokine response. This result is consistent with previous studies showing that CNs have an immune-enhancing effect [10, 11].

Detailed bioinformatic analysis of DEGs suggested that genes associated with cell viability of neurons, epithelial cells and muscle cells were down-regulated at 1hpi in CNs-Mdh-immunized group as compared to other immunized group. Then, genes associated with active

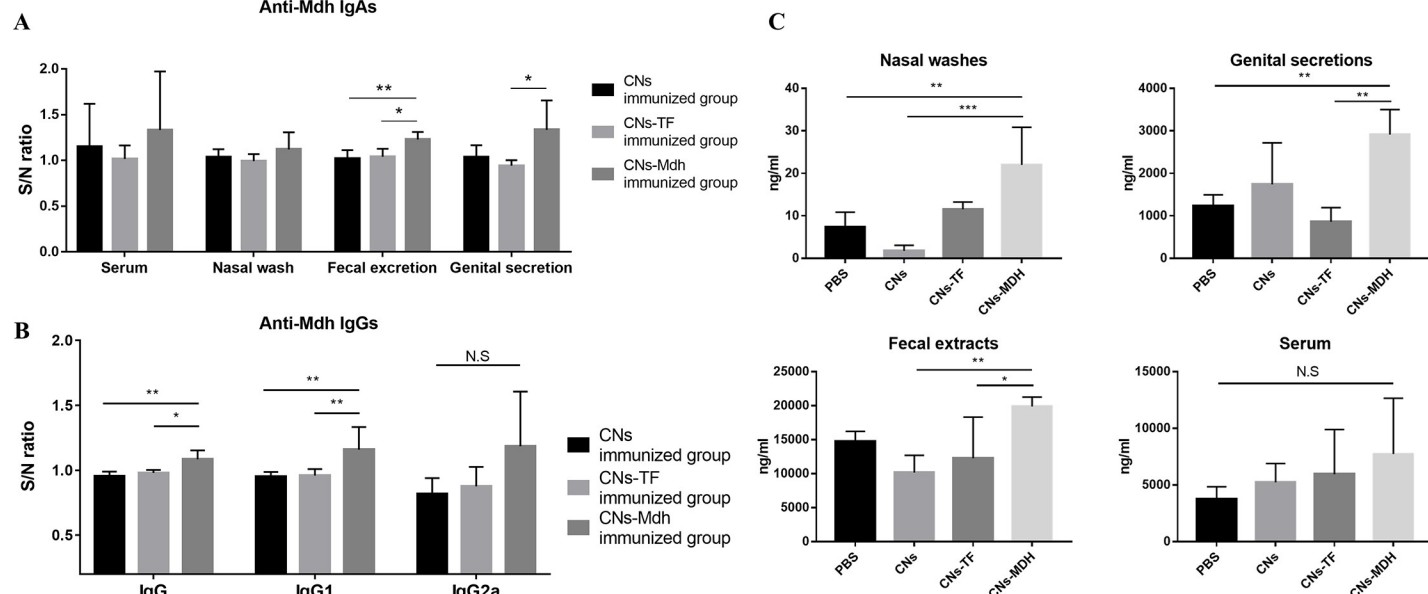

**Fig 7. Antibody measurements at 2 weeks after primary intranasal immunization.** (A) Specific IgA antibodies in sera, nasal wash, fecal extract, and genital secretions. Significant production of IgA was detected in the fecal excretions and genital secretions from the CNs-Mdh-immunized group compared to CNs-TF-immunized group. (B) Specific IgG antibodies at 2 wpi. The CNs-Mdh elicited a significantly higher titer of specific IgG than that of the CNs-TF-immunized group which served as the vector control. The main subtype produced after immunization was IgG1, while the titer of IgG2a was not significantly enhanced following immunization. (C) Total IgA antibodies in sera, nasal wash, fecal extract, and genital secretions. Titers of IgA in genital secretions and fecal excretions were significantly increased in CNs-Mdh-immunized group compared with those following the CNs-TF-immunization. In the nasal washes, a high titer of IgA in the CNs-Mdh-immunized group was measured, indicating significance compared to the PBS- and CNs-immunized groups. However, there were no significant differences in sera. Groups were statistically compared using one-way ANOVA with Tukey's post hoc multiple comparison test. The results for specific antibodies are expressed as the sample to negative control (PBS) ratio (S/N ratio).

movement of immune cells including T cell, leukocytes, phagocytes and granulocytes in NALT were gradually up-regulated at 6-12hpi, indicating that the antigen recognition was activated. In addition, pathway analysis showed that the DEGs were significantly enriched in multiple inflammatory pathways including "IL-6 Signaling" and "HMGB1 Signaling" in CNs-Mdh-immunized group.

IL-6 has been indicated as a key cytokine to the control of Th1/Th2 differentiation [30]. Cellular response (Th1) is characterized by production of IgG2a and known to be crucial to overcome the bacterial infection [31]. Humoral response (Th2), which mainly produce IL-4, IL-5, IL-6 and IL-10, is also characterized by production of IgG1 [30, 32]. Protective immunity of host against *B. abortus* is likely to be mediated by complicated balance between Th1 and Th2 related immune responses. A recent study showed that IL-6 contribute to resistance against *B. abortus* in macrophages and CD8+ T cell differentiation, priming the cellular response during the *Brucella* infection [33]. In addition, the antibody profile of regimens vaccinated with the live attenuated vaccine strains for brucellosis, S19 and RB51, were reported to be predominantly IgG1 whereas they are known to induce strong cellular immunity [34]. Although DEGs from CNs-Mdh vs CNs-TF showed that Th17 pathway were potentially activated at 6 hpi, activation of IL-6 signaling pathway was more statistically significant. At the 2 wpi, the production of specific IgG1 and IgG2a in sera indicates that a mixed Th1-Th2 response were elicited by CNs-Mdh. In addition, intranasal immunization of CNs-Mdh induced systemic mucosal immunity with Th2-related response at 6 wpi in our previous study. The CD4+ T cells from NALT of naïve mice were reported to be Th0 cells, suggesting that T helper subsets could be differentiated into the Th1, Th2, and Th17 cells within NALT

correspondingly to the antigen [35]. Taken together, these results suggest that immunocompetent cells in NALT activated IL-6 signaling pathway, elicited a mixed Th1-Th2 repsonse in early stages and then finally differentiated into Th2 by CNs-Mdh.

Recent findings have highlighted that HMGB1 pathway is crucial not only for the inducing inflammatory responses of upper respiratory tract (URT) but also for mediating DCs maturation and activation [36–39]. HMGB1 is a archetypical alarmin that can act as a danger signal that initiates the host defense response [40]. HMGB1 signaling was suggested as a immunostimulatory signal that induces DC maturation and T-cell-mediated immunity in rat [39]. Shimizu et al. showed that nasal secretions contain substantial amounts of HMGB1 and TNF-α stimulation induces the production of HMGB1, which, in turn, upregulates the production of IL-6 and IL-8 by nasal epithelial cells via TLR 4 [38]. In our results, HMGB1 pathway via TLR4 as well as IL-6 signaling and DCs maturation were predicted to be activated in CNs-Mdh-immunized group. Another study showed that oral administration of a representative mucosal adjuvant, cholera toxin, triggers the release of HMGB1 from damaged intestinal epithelial cells, and that the released HMGB1 may mediate the activation of mucosal DCs, cytotoxic T lymphocytes and IgA production in the intestine [37]. Although the relationship between innate immune pathway and production of IgA is still not well defined, we reasonably assume that activated HMGB1 signaling pathways with IL-6 signaling and DCs maturation are shown by CNs-Mdh due to the relevance of followed production of IgA.

Secretory IgA-mediated mucosal immunity is not always induced through mucosal immunization [41]. To produce IgA, homing of IgA-ASCs in the mucosa is required. DCs in NALT present intranasally administered antigens to naive B lymphocytes, by expressing type 1 retinoic acid, nitric oxide (NO) and proliferation-inducing ligand [20–23]. In addition, when B cells are activated by T cell-dependent antigens that elicit helper CD4 T-cell responses, they become either ASCs [22, 23]. According to the transcriptomic analysis, genes related in activation of lymphocytes, chemotaxis and cell movement of DCs were actively expressed at NALT following CNs-Mdh immunization. Gene networks also showed that expression of *Il1a*, *Il1b*, *Tnf*, *Vip*, and *Csf2rb* connected these networks. Additionally, genes involved in synthesis of retinoic acid (*Aldh1a3*) [42], modulators for T cell and DC function (*Csf*, *Csf1*, and *Csf3*) [43], generation of IgA-ASCs and IgA (*Tgfb1*, *Tgfbi*, and *Cd40*) [44] were expressed concomitantly. In the ELISA results, significant titers of antigen specific IgA in genital secretions and fecal excretion and total IgA in nasal wash, genital secretions and fecal excretion were observed in the CNs-Mdh immunized group at 2 wpi. These activated pathways and gene networks may have led to the statistically difference in titer of systemic IgA.

Collectively, intranasal immunization of CNs-Mdh triggered differentially gene expression associated with cellular movement of immune cells, IL-6 signaling pathway, HMGB1 signaling pathway and DCs maturation within NALT. In addition, significantly increased amount of systemic IgA and a mixed Th1-Th2 response were elicited in CNs-Mdh immunized group at 2wpi. These results suggest that *B. abortus* Mdh-loaded CNs drives the IL-6 signaling and HMGB1 signaling pathway followed by DCs maturation in URT, and the significantly increased production of antigen-specific IgA and IgG.

## Supporting information

**S1 Checklist. The ARRIVE guidelines checklist.**
(PDF)

**S1 Fig. Genes and functions that commonly regulated between CNs-Mdh vs CNs and CNs-TF vs CNs.** (A) Venn diagram of DEGs from only the CN-Mdh- and CN-TF-immunized groups compared to the CNs-immunized group. (B) Gene Ontology enrichment analysis

using commonly regulated genes from only the CN-Mdh and CN-TF-immunized groups.
(TIF)

**S2 Fig. Correlation of gene expression values between RNA-seq and qRT-PCR.** The six validated genes (*Il1rn*, *Batf*, *Tlr4*, *Il6*, *Il23a* and *Tnf*) were important for the inflammatory response and differentiation of Th17 cells.
(TIF)

**S1 Raw image.**
(PDF)

**S1 Table. Primers used for qRT-PCR.**
(DOCX)

## Acknowledgments

We thank Hobin Lee for help with making nanoparticles. We thank The National Instrumentation Center for Environmental Management (NICEM) for help with the SEM and DLS analyses. We thank TERAGEN for RNA-seq.

## Author Contributions

**Conceptualization:** Soojin Shim, Young Bin Im, Han Sang Yoo.

**Data curation:** Soojin Shim, Sang Hee Soh, Young Bin Im, Choonghyun Ahn, Hyun-Eui Park, Woo Bin Park, Suji Kim, Han Sang Yoo.

**Formal analysis:** Soojin Shim, Sang Hee Soh, Hong-Tae Park, Woo Bin Park, Suji Kim.

**Funding acquisition:** Han Sang Yoo.

**Investigation:** Young Bin Im, Hong-Tae Park, Hyun-Eui Park, Han Sang Yoo.

**Methodology:** Soojin Shim, Sang Hee Soh, Young Bin Im, Hyun-Eui Park, Woo Bin Park, Suji Kim.

**Project administration:** Han Sang Yoo.

**Resources:** Soojin Shim, Sang Hee Soh, Woo Bin Park, Suji Kim.

**Supervision:** Han Sang Yoo.

**Validation:** Choonghyun Ahn, Hong-Tae Park, Hyun-Eui Park, Han Sang Yoo.

**Visualization:** Hong-Tae Park, Han Sang Yoo.

**Writing – original draft:** Soojin Shim, Choonghyun Ahn.

**Writing – review & editing:** Soojin Shim, Choonghyun Ahn, Han Sang Yoo.

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
