## [Decision Letter · Decision Letter 0]

5 Dec 2019

PONE-D-19-28863

Induction of Systemic Immunity Through Nasal-associated Lymphoid Tissue (NALT) of Mice Intranasally Immunized with Brucella abortus malate dehydrogenase-loaded Chitosan Nanoparticles

PLOS ONE

Dear Dr. Yoo,

Thank you for submitting your manuscript to PLOS ONE. After careful consideration, we feel that it has merit but does not fully meet PLOS ONE’s publication criteria as it currently stands. Therefore, we invite you to submit a revised version of the manuscript that addresses the points raised during the review process.

Your manuscript has been reviewed by an expert in your field. Based on the critiques from the reviewer, a  major revision is requirement before a final decision can be made.

We would appreciate receiving your revised manuscript by 4 weeks. To enhance the reproducibility of your results, we recommend that if applicable you deposit your laboratory protocols in protocols.io, where a protocol can be assigned its own identifier (DOI) such that it can be cited independently in the future. For instructions see: http://journals.plos.org/plosone/s/submission-guidelines#loc-laboratory-protocols

We look forward to receiving your revised manuscript.

Kind regards,

Yung-Fu Chang

Academic Editor

PLOS ONE

Journal Requirements:

2. As part of your revision, please complete and submit a copy of the ARRIVE Guidelines checklist, a document that aims to improve experimental reporting and reproducibility of animal studies for purposes of post-publication data analysis and reproducibility: https://www.nc3rs.org.uk/arrive-guidelines. Please include your completed checklist as a Supporting Information file. Note that if your paper is accepted for publication, this checklist will be published as part of your article.

3. We note that you are reporting an analysis of a microarray, next-generation sequencing, or deep sequencing data set. PLOS requires that authors comply with field-specific standards for preparation, recording, and deposition of data in repositories appropriate to their field. Please upload these data to a stable, public repository (such as ArrayExpress, Gene Expression Omnibus (GEO), DNA Data Bank of Japan (DDBJ), NCBI GenBank, NCBI Sequence Read Archive, or EMBL Nucleotide Sequence Database (ENA)). In your revised cover letter, please provide the relevant accession numbers that may be used to access these data. For a full list of recommended repositories, see http://journals.plos.org/plosone/s/data-availability#loc-omics or http://journals.plos.org/plosone/s/data-availability#loc-sequencing.

Additional Editor Comments (if provided):

Reviewers' comments:

Reviewer's Responses to Questions

**Comments to the Author**

1. Is the manuscript technically sound, and do the data support the conclusions?

Reviewer #1: Partly

2. Has the statistical analysis been performed appropriately and rigorously? 

Reviewer #1: Yes

3. Have the authors made all data underlying the findings in their manuscript fully available?

Reviewer #1: Yes

4. Is the manuscript presented in an intelligible fashion and written in standard English?

Reviewer #1: Yes

5. Review Comments to the Author

Reviewer #1: Overall, the manuscript is well written and the experimental design is straightforward and easy to understand. However, the authors are over-reaching in their conclusions based on the available data and there are some immunological concepts that need to be revised throughout the manuscript. The authors should consider two key points: 1) Chitosan has natural adjuvant activity in that it induces dendritic cell activation to promote TH1 responses, mediated through STING, which doesn’t quite fit with the authors conclusion of a TH2-type of response, nor does the data really support the conclusion that a TH2 response has been induced; and 2) For Brucella clearance, induction of a TH1 response is necessary, so induction of a TH2 response in not a desirable outcome from a vaccine against brucellosis. The authors should consider this and address it in their discussion.

Below are some of the primary concerns regarding immunological concepts throughout the manuscript:

Line 306-307: “….active movement of leukocytes, phagocytes, granulocytes….indicating that the antigen recognition was activated.” Migration of these cells is not indicative of antigen recognition, but rather inflammation. Sentence should be revised.

Line 329-330. Referring to IL-1beta, IL-6, NF-kB, etc. as inducers of Th2 responses needs revision. The mentioned molecules are indeed involved in the proinflammatory response, but not necessarily specific drivers of Th2 differentiation.

Line 439-440: “IL-6 has been indicated as a key cytokine for the differentiation of Th2 cells in both mice and humans.” IL-6 is involved in both TH1 and TH2 responses, and at the early stages that these samples are being analyzed, the skewing of adaptive immune responses cannot be determined. Sentence should be revised.

Line 480-482: This is overinterpretation of the data. The work simply shows that upregulation of HMGB1, IL-6, and DC maturation at the early time points after immunization may be correlated to increased IgA production at 2 wpi. These experiment were not designed to show causation.

The IgG and IgA data is not very impressive. While there is statistical significance, the levels of total IgG, IgG1 and IgG2a are not very high when compared to the rest of the experimental groups. Perhaps looking at later time points would have helped enhance that response. Additionally, the antibody data does not support the induction of a TH2 response, rather, it appears mixed. Other parameters such as cytokine production from recall responses would have been a better approach to demonstrate the type of immune response that was induced by this type of immunization.

It should be noted that there is no mention of antibody data in the introduction of the manuscript, yet the authors utilize the antibody data to tie in their transcriptomic data to an immunological phenotype. It is in the trying connect these two pieces of data (i.e. gene expression and humoral responses) that the manuscript loses its strength, as these data are not congruent with one another.

6. PLOS authors have the option to publish the peer review history of their article (what does this mean?). If published, this will include your full peer review and any attached files.

Reviewer #1: No

---

## [Author Response · Author response to Decision Letter 0]

10 Dec 2019

Comments from the editor and reviewers:

I would like to thank for the comments from reviewer. Comments from reviewer make improve our manuscript. We revised our manuscript based the comments and tried our best to revise. 

- Reviewer's comments:

Reviewer #1

1. Overall, the manuscript is well written and the experimental design is straightforward and easy to understand. However, the authors are over-reaching in their conclusions based on the available data and there are some immunological concepts that need to be revised throughout the manuscript. The authors should consider two key points: 1) Chitosan has natural adjuvant activity in that it induces dendritic cell activation to promote TH1 responses, mediated through STING, which doesn’t quite fit with the authors conclusion of a TH2-type of response, nor does the data really support the conclusion that a TH2 response has been induced; and 2) For Brucella clearance, induction of a TH1 response is necessary, so induction of a TH2 response in not a desirable outcome from a vaccine against brucellosis. The authors should consider this and address it in their discussion.

Answer. 

We appreciate your valuable comment. As your opinion, we might have jumped to the conclusion. In our previous study, intranasal immunization of CNs-Mdh induced systemic mucosal immunity with Th2-related response at 6 wpi [1]. In this study, we aimed to observe whether our antigens properly activate the NALT, the target site of the immunization, and the antibody is produced after intranasal immunization of CNs-Mdh. The pathway analysis showed that the DEGs were significantly enriched in multiple inflammatory pathways including “IL-6 Signaling”, "Dendritic cells maturation" and “HMGB1 Signaling” in CNs-Mdh-immunized group. At the 2 wpi, the production of specific IgG1 and IgG2a in sera indicates that a mixed Th1-Th2 response were elicited by CNs-Mdh. Taken together, we concluded that immunocompetent cells in NALT activated IL-6 signaling pathway and DCs maturation in early time point and elicited a mixed Th1-Th2 response in early stages, then finally differentiated into Th2 by CNs-Mdh. Therefore, we revised the concept of the manuscript (Lines 451-459).

Until now, the vaccines inducing cellular immunity have been known to be effective for intracellular bacteria. However, the subunit vaccines which induce not only humoral immunity but also complex immune response are being suggested as a replacement to the existing vaccines recently [2]. In addition, protective immunity of host against Brucella abortus is mediated by both cellular and humoral immune responses. A recent study showed that IL-6 promotes Brucella abortus clearance in macrophages and CD8+ T cell differentiation, priming the Th1 response during Brucella infection [3]. The live attenuated vaccine strains for brucellosis, S19 and RB51 are known to induce strong cellular immunity, but the antibody profile observed in both regimens were reported as predominantly IgG1 [2]. This discrepancy suggests that the course of infection and immune protection against Brucella involves complicated balance between Th1 and Th2 related immune response. 

We tried to suggest not only the new antigen delivery system, but also a convenient method to induce systematic IgG and IgA immune responses with relatively small amount of Brucella antigen. In addition, a functional assessment of this antibody is planned for further study by modulating infection. We believe that this study could contribute to the development of subunit vaccine. Finally, the sentences about protective immunity against Brucella were added in the discussion (Lines 443-449). Thank you.

2. Line 306-307: “….active movement of leukocytes, phagocytes, granulocytes….indicating that the antigen recognition was activated.” Migration of these cells is not indicative of antigen recognition, but rather inflammation. Sentence should be revised.

Answer. 

The sentence has been amended as recommended (Lines 309-310). 

3. Line 329-330. Referring to IL-1beta, IL-6, NF-kB, etc. as inducers of Th2 responses needs revision. The mentioned molecules are indeed involved in the proinflammatory response, but not necessarily specific drivers of Th2 differentiation.

Answer. 

The sentence has been amended as recommended (Lines 330-332).

4. Line 439-440: “IL-6 has been indicated as a key cytokine for the differentiation of Th2 cells in both mice and humans.” IL-6 is involved in both TH1 and TH2 responses, and at the early stages that these samples are being analyzed, the skewing of adaptive immune responses cannot be determined. Sentence should be revised.

Answer. 

The sentence has been amended as recommended (Line 440).

5. Line 480-482: This is overinterpretation of the data. The work simply shows that upregulation of HMGB1, IL-6, and DC maturation at the early time points after immunization may be correlated to increased IgA production at 2 wpi. These experiment were not designed to show causation.

Answer. 

We appreciate your valuable comment. As the relationship between innate immune pathway and production of IgA is still not well defined, we found other studies about HMGB1 signaling, IL-6 signaling, DCs maturation and genes related to production of IgA as we described in the manuscript (Lines 465-487). Since the secretory IgA-mediated mucosal immunity is not always induced through mucosal immunization at local site, we thought that activated pathways and gene network induced by CNs-Mdh were related to production of IgA. However, as your opinion, we have not examined causative relationship and we revised the sentence (Lines 489-490). Thank you.

6. The IgG and IgA data is not very impressive. While there is statistical significance, the levels of total IgG, IgG1 and IgG2a are not very high when compared to the rest of the experimental groups. Perhaps looking at later time points would have helped enhance that response. Additionally, the antibody data does not support the induction of a TH2 response, rather, it appears mixed. Other parameters such as cytokine production from recall responses would have been a better approach to demonstrate the type of immune response that was induced by this type of immunization.

Answer. 

We appreciate for your valuable comment. As we described above, we revised the concept of the manuscript. In addition, here is the reason we looked the early time point; since we observed humoral responses at 4wpi and 6 wpi in our previous study [1], we aimed to observe whether our antigens properly activate the NALT and the antibody is produced after intranasal immunization of CNs-Mdh in this study. We regarded 2 weeks long enough, considering the production of IgG and IgA after antigen recognition in NALT and inducing systemic response. It would be more helpful to observe enough cytokines produced by recall responses, but considering high prices of RNA-seq and the number of samples, we had to selectively observe the transcriptomic data and antibody. I hope your understanding on our current limitation on resources.

7. It should be noted that there is no mention of antibody data in the introduction of the manuscript, yet the authors utilize the antibody data to tie in their transcriptomic data to an immunological phenotype. It is in the trying connect these two pieces of data (i.e. gene expression and humoral responses) that the manuscript loses its strength, as these data are not congruent with one another.

Answer. 

We appreciate for your valuable comment. We added sentence explaining the reason we have investigated the titer of antibody in introduction (lines 68-72; 80-81).

[References]

1. Soh SH, Shim S, Im YB, Park HT, Cho CS, Yoo HS. Induction of Th2-related immune responses and production of systemic IgA in mice intranasally immunized with Brucella abortus malate dehydrogenase loaded chitosan nanoparticles. Vaccine. 2019;37(12):1554-64.

2. Dorneles EM, Lima GK, Teixeira-Carvalho A, Araujo MS, Martins-Filho OA, Sriranganathan N, Qublan H, Heinemann MB, Lage AP, 2015. Immune Response of Calves Vaccinated with Brucella abortus S19 or RB51 and Revaccinated with RB51. PLOS one 10, e0136696.

3. Hop HT, Huy TXN, Reyes AWB, Arayan LT, Vu SH, Min W, Lee HJ, Kang CK, Kim DH, Tark DS, Kim S, 2019. Interleukin 6 (IL-6) promotes Brucella abortus clearance by controlling bactericidal activity of macrophages and CD8(+) T cell differentiation. Infect Immun. 18;87(11).

---

## [Decision Letter · Decision Letter 1]

16 Jan 2020

Induction of Systemic Immunity Through Nasal-associated Lymphoid Tissue (NALT) of Mice Intranasally Immunized with Brucella abortus malate dehydrogenase-loaded Chitosan Nanoparticles

PONE-D-19-28863R1

Dear Dr. Yoo,

We are pleased to inform you that your manuscript has been judged scientifically suitable for publication and will be formally accepted for publication once it complies with all outstanding technical requirements.

With kind regards,

Yung-Fu Chang

Academic Editor

PLOS ONE

Additional Editor Comments (optional):

Reviewers' comments:

Reviewer's Responses to Questions

**Comments to the Author**

1. If the authors have adequately addressed your comments raised in a previous round of review and you feel that this manuscript is now acceptable for publication, you may indicate that here to bypass the “Comments to the Author” section, enter your conflict of interest statement in the “Confidential to Editor” section, and submit your "Accept" recommendation.

Reviewer #1: All comments have been addressed

2. Is the manuscript technically sound, and do the data support the conclusions?

Reviewer #1: Yes

3. Has the statistical analysis been performed appropriately and rigorously? 

Reviewer #1: Yes

4. Have the authors made all data underlying the findings in their manuscript fully available?

Reviewer #1: Yes

5. Is the manuscript presented in an intelligible fashion and written in standard English?

Reviewer #1: Yes

6. Review Comments to the Author

Reviewer #1: (No Response)

7. PLOS authors have the option to publish the peer review history of their article (what does this mean?). If published, this will include your full peer review and any attached files.

Reviewer #1: No

---

## [Editor Report · Acceptance letter]

21 Jan 2020

PONE-D-19-28863R1 

Induction of Systemic Immunity Through Nasal-associated Lymphoid Tissue (NALT) of Mice Intranasally Immunized with *Brucella abortus* malate dehydrogenase-loaded Chitosan Nanoparticles 

Dear Dr. Yoo:

I am pleased to inform you that your manuscript has been deemed suitable for publication in PLOS ONE. Congratulations! Your manuscript is now with our production department. 

With kind regards,

on behalf of

Dr. Yung-Fu Chang 

Academic Editor

PLOS ONE